# CODECLOAK: A METHOD FOR MITIGATING CODE LEAKAGE BY LLM CODE ASSISTANTS

## ABSTRACT

LLM-based code assistants are becoming increasingly popular among developers. These tools help developers improve their coding efficiency and reduce errors by providing real-time suggestions based on the developer's codebase. While beneficial, the use of these tools can inadvertently expose the developer's proprietary code to the code assistant service provider during the development process. In this work, we propose a method aimed at mitigating the risk of code leakage when using LLM-based code assistants. CodeCloak is a novel deep reinforcement learning agent that manipulates the prompts before sending them to the code assistant service. CodeCloak aims to achieve the following two contradictory objectives: (i) minimizing code leakage, while (ii) preserving relevant and useful suggestions for the developer. Our evaluation, employing StarCoder and Code Llama, LLM-based code assistants models, demonstrates CodeCloak's effectiveness on a diverse set of code repositories of varying sizes, as well as its transferability across different models. We also designed a method for reconstructing the developer's original codebase from code segments sent to the code assistant service (i.e., prompts) during the development process, to thoroughly analyze code leakage risks and evaluate the effectiveness of CodeCloak under practical development scenarios.

## 1 INTRODUCTION

**LLM-Based Code Assistants.** The use of large language models (LLMs) for code generation is growing and becoming increasingly useful to the global community Vaithilingam et al. (2022). The efficiency of these models has been demonstrated in a number of studies Chang et al. (2023). Code assistance tools, like Code Llama,[1] GitHub Copilot,[2] and StarCoder,[3] leverage advanced LLMs to provide real-time suggestions, improving developers' coding efficiency and reducing errors. The flow of interaction with a code assistant during the coding process takes place as follows: First, the developer writes code or describes the desired functionality in code or free text, within an integrated development environment (IDE). The code assistant client, which is often a plugin integrated in the IDE, actively monitors the code being developed. It gathers relevant contextual information, including the developer's recent interactions in the IDE and segments of code that may be relevant from other code sections and files in order to generate code suggestions. Then, based on the data gathered, the code assistant client generates a prompt that contains relevant information for generating useful suggestions and sends it to the code assistant service (also referred to as the model). The code assistant service uses an LLM to generate suggestions about how the code can be completed and sends them back to the code assistant client. Finally, these suggestions are presented to the developer via the IDE interface. The developer can integrate suitable suggestions in the code.

**Privacy and Security Risks.** While LLMs have become invaluable in various tasks, their integration and use may inadvertently introduce security risks and expose sensitive data. In the context of code assistants, the use of LLMs may introduce new vulnerabilities to the developed code Pearce et al. (2022) or unintentionally expose sensitive data (i.e., proprietary code). Previous studies have addressed three main security and privacy concerns: (1) the risk of inferring whether specific code was used to train an LLM model, known as a membership inference attack Carlini et al. (2021); Sun et al. (2022); Niu et al. (2023); (2) the challenge of preventing proprietary code from being recommended by an LLM-based code assistant, particularly when the model has been trained on similar

---

[1] https://ai.meta.com/blog/code-llama-large-language-model-coding/
[2] https://github.com/features/copilot
[3] https://huggingface.co/bigcode/starcoder

code, such as code from an open-source repository, by employing code manipulation strategies before the code is released Ji et al. (2022); and (3) exploration of whether an LLM code assistant can be poisoned during its trainingphase, leading to the presentation of intentional suggestions of vulnerable code to developers, thereby introducing vulnerabilities into the developed software Perry et al. (2022); Khoury et al. (2023); Inan et al. (2021). However, while these security and privacy concerns have received research and media attention[4], to the best of our knowledge, the risk and potential extent of private code leakage via prompts transmitted to a third-party code assistant service, along with strategies to mitigate this risk, have not been investigated. We regard this as a significant risk given that the entire codebase of an organization's proprietary software could be leaked to an untrusted third-party service provider or an adversary capable of intercepting communication between developers and code assistant services. One approach for addressing the private code leakage concern is to use a local code assistant LLM, which is possible due to the proliferation of open-source models. The use of local models prevents sending prompts outside the network. However, commercial remote alternatives offer advantages in terms of model freshness, maintenance, performance, ease of use, and integration with well-designed plugins for IDEs. The convenience of these commercial services continues to attract developers, despite the privacy concerns associated with remote services.

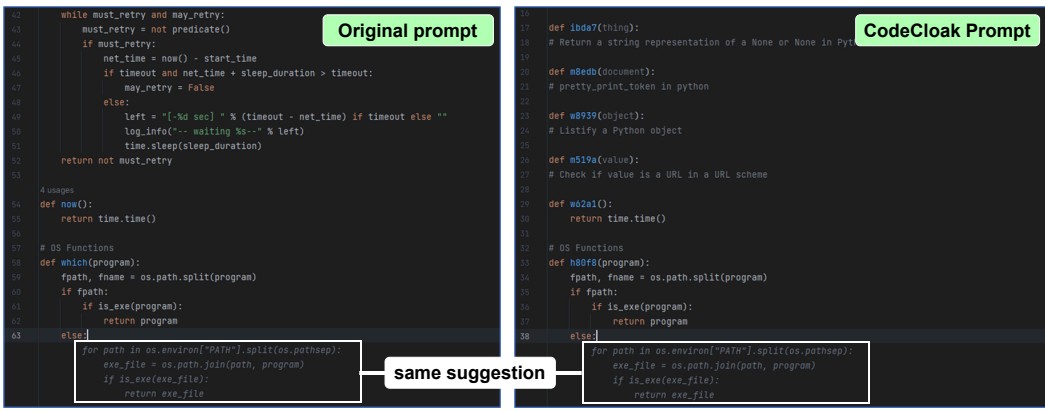

Figure 1: An example of CodeCloak's performance. Left: an original prompt and the code suggested by StarCoder. Right: the manipulated prompts obtained by applying CodeCloak and the code suggested by StarCoder for the manipulated prompt.

**Goal.** There is a lack of comprehensive methods that effectively mitigate code exposure. In this research, we address this gap by proposing CodeCloak, a novel deep reinforcement learning (DRL) agent that manipulates code prompts before they are sent to the code assistant service. CodeCloak addresses the critical and underexplored issue of code leakage in LLM-based code assistants. By defining and formalizing the leakage problem in real-time IDE environments, we provide a foundation for enhancing privacy when using code assistants. Our approach solves this challenge by manipulating prompts in real-time to mitigate leakage without requiring modifications to the underlying assistant model, ensuring both practicality and seamless integration into existing workflows.

As an example, in Figure 1 we present an original prompt (left) and a manipulated prompt generated by CodeCloak (right), both provided as input to the StarCoder code suggestion model. At the bottom, the code suggestions provided by StarCoder for the original prompt and manipulated prompt are presented (outlined in white). As can be seen, while the manipulated prompt is significantly different from the original one, the response in both cases (suggested code) is similar. In our evaluation, our mitigation method, CodeCloak, was shown to successfully reduce code leakage to an average of 40% while maintaining an average of 70% similarity in terms of the CodeBLEU metric, between the suggestions obtained for the manipulated prompts and those for the original prompts.

**Contributions.** The contributions of our work can be summarized as follows: (1) To the best of our knowledge, we are the first to explore the risk of code leakage via the prompts and propose a mitigation. (2) We present CodeCloak, a novel method based on a DRL agent, which is aimed at both

---

[4]https://www.forbes.com/sites/siladityaray/2023/05/02/samsung-bans-chatgpt-and-other-chatbots-for-employees-after-sensitive-code-leak/?sh=581383d16078

reducing code leakage and preserving useful suggestions. CodeCloak is generic and transferable and thus, can be applied to any LLM code assistant. This approach can also be extended to LLMs in other domains. (3) We present a method for reconstructing codebase from code segments; in our usecase, we utilize prompts sent from the IDE to the code assistant model. We utilize the proposed method in order to evaluate the amount and quality of the potential code leakage. and (4) We provide to the research community a unique dataset containing authentic requests (prompts) and responses (suggestions) generated by simulating the code development process and interactions with code assistant service. This dataset captures the prompts sent to the code assistant services along with the corresponding suggestions received, enabling evaluation of code leakage and mitigation techniques.

## 2  RELATED WORK

In this section, we provide a review of related work relevant to our study, including work on: privacy and security risks inherent in LLM code assistants, and prompt editing.

**Privacy and Security Risks.** As the adoption of code assistant tools has increased, inherent security and privacy risks have emerged Sandoval et al. (2023); Pearce et al. (2022).

Pearce et al. (2022) investigated the security implications of GitHub Copilot's code contributions by designing scenarios that crafted prompts sent to Copilot results in vulnerable code being suggested. Incorporating suggestions from code assistants can intentionally or unintentionally introduce vulnerabilities or malicious content into the developer's software. Inan et al. (2021) demonstrated how improperly use of these tools can perpetuate existing security flaws in their training data. Khoury et al. (2023) evaluated 21 programs in various programming languages, focusing on their vulnerability to common security flaws. There are also concerns regarding inference attacks, when code repositories are used to train LLMs. The risk of inferring whether specific code was used to train the LLM model, also know as a membership inference attack (MIA), was discussed in Carlini et al. (2021); Sun et al. (2022); Niu et al. (2023); Lukas et al. (2023). For example, Lukas et al. (2023) proposed a novel MIA methodology and evaluated it on multiple code generation LLMs. They designed specific prompts to induce personally identifiable information (PII) leakage in programming languages and established a practical, automated pipeline for verifying such leaks. To mitigate the privacy risks associated with unauthorized learning and code leakage, several studies have proposed different strategies to protect sensitive information from LLM-based code assistants. Ji et al. (2022) addressed the challenge of protecting proprietary code against unauthorized learning by LLM-based code assistants, particularly when these models are trained on open-source repositories. They proposed a method that involves applying lightweight, semantics-preserving transformations to the code before it is open-sourced. This strategy effectively obstructs models trained on the modified code, safeguarding the code from unauthorized learning while preserving its readability and functionality. Lin et al. (2024) suggested a method defense that converts sensitive information in the original LLM input prompt into an encrypted format that, while indecipherable to human interpretation, retains its informativeness for the LLM to process. Pape et al. (2024) introduced prompt obfuscation to prevent the extraction of the system prompt using Greedy Coordinate Gradient. While Those researches focused on mitigating leakage for general LLMs, we focus on code LLMs. Patil et al. (2023) presented a model level defense, which deletes sensitive information from the LLM model weights directly, assuming access to the model parameters, while we assume black-box access to the LLM. While the studies above focused on assessing the security and potential vulnerabilities of code suggested by the code assistant, or developing methods for protecting proprietary code, we focus on the risks of code leakage via prompts when using code assistance models.

**Prompt Editing.** On the rise of pretrained generative language models, the domain of prompt engineering for these LLMs has rapidly gained traction. Consequently, there is a growing focus among researchers on prompt editing to refine and optimize input prompts to enhance model output quality and alignment with user intentions Wang et al. (2023); Hertz et al. (2022). Several studies have explored prompt editing techniques using RL. For example, Zhang et al. (2022) presented TEMPERA, an RL method for dynamically editing prompts to enhance LLM performance on NLP tasks (such as sentiment analysis and topic classification). Kong et al. (2024) introduced PRewrite, an automated prompt engineering method using RL for prompt optimization. Our study introduces a data level defense that combines RL methods along with prompt editing, in order to mitigate the code leakage during the code writing (development) process. Whereas previous studies mainly focused on general LLMs in different phases of model deployment, our defense specifically aims at code assistant provider. Our methodology can be integrated into diverse LLM code assistant models, without access to the model itself or its parameters.

# 3 CODECLOAK: CODE LEAKAGE MITIGATION METHOD

## 3.1 PROBLEM DEFINITION

As mentioned in Section 1, to receive relevant suggestions, code assistant services rely on developers to provide the code they are currently working on. Let $P$ be the sequence of prompts sent from the developer's IDE to the code assistant service during the development process. Correspondingly, let $S$ be the sequence of suggestions received from the code assistant service for the prompts $P$. Our goal is to achieve two seemingly contradictory objectives simultaneously: *(i)* minimize the exposure and leakage of the developer's proprietary code by changing and manipulating the prompts $P$ before sending them to the code assistant service, and *(ii)* preserve the relevance and usefulness of the suggestions the developer receives for the manipulated prompts, by ensuring that they are as similar as possible to $S$. In our research, we treat all code sent to the LLM-based assistant as private or sensitive. This includes not only proprietary code but also open-source code that may inadvertently reveal the functionality of the developed application or the use of vulnerable code and libraries.

In the context of code assistants, we observe that the prompts provided to the code assistant often contain excessive information that may not be essential for generating meaningful and relevant code suggestions. This observation serves as the basis of our proposed method. To leverage this insight, we introduce a trained Deep Reinforcement Learning (DRL) model, which is tasked with identifying information that is not essential or has a negligible effect on the suggestions' quality for a given prompt, and manipulate the prompt accordingly.

The sequential and context-sensitive nature of prompt manipulation makes reinforcement learning (RL) particularly suitable for this task: (1) Sequential Decision-Making - Dynamically adjust manipulations as each modification affects future decisions. (2) Learning and Planning - Understand the effects of manipulations on code completions and plan optimal sequences of actions. (3) Practical Deployment: Operate efficiently in real-time environments, thanks to RL's lightweight and resource-efficient nature compared to alternatives like fine-tuning LLMs.

By leveraging RL, our approach ensures that developer's code is adequately modified, minimizing its exposure while maintaining the relevance and usefulness of the code assistant's suggestions. Background on DRL can be found in the Appendix B.

## 3.2 CODECLOAK DESCRIPTION

As illustrated in Figure 2, CodeCloak is placed between the IDE and the code assistant service, intercepting and modifying the prompts to reduce code exposure while preserving the relevance of the assistant's suggestions. It can run locally on the developer's machine or in the organization's gateway. The prompt manipulation process is modeled as a Markov decision process (MDP), where the agent learns to select the optimal actions (the manipulations) in order to maximize a reward that aims to *minimize code leakage* and *maintain suggestion relevance*. The key components of the DRL agent are described in the subsections below:

### 3.2.1 STATES

To enable the agent to effectively choose the appropriate manipulation, we provide it with detailed information about the prompt it is currently processing. In each iteration, the agent determines which manipulation to apply by analyzing the state that captures this information. We incorporate the following key elements into the state representation:

**Prompt Embedding.** The primary element is the prompt that is being manipulated. Since prompts are textual data, we need to represent them in a way that captures their semantic meaning and context. To achieve this, we feed the prompt to a pretrained code encoder model and use the last hidden state as an embedding representation of the prompt. This embedding technique allows the agent to capture the prompt's semantic information, which is essential for making informed decisions. In cases where the prompt length exceeds the encoder's maximum token limit, we split the prompt into segments. Each segment is processed individually through the encoder to generate its embedding and is represented as a separate state. This segmentation approach allows the agent to handle longer prompts effectively, providing better context for targeted modifications. Moreover, this method is efficient and tailored for real-time use, ensuring that manipulations are applied within the short time constraints of practical coding scenarios.

**Relative Location.** In the state, we include a fragment that represents the segment's relative location within the full prompt. This positional information is important for the agent ,enabling it to better understand the order and placement of the code segments within the full prompt.

**Cursor Position.** The prompt sent to the code assistant includes the developer's cursor location, helping the code assistant understand where in the code the developer is currently focusing on. In the state, we also include the relative locations of the developer's cursor position (divided into two fragments: one indicates the cursor's relative line location, and the second indicates the relative location in the line). Including the cursor position in the state representation is important, since it provides contextual information about the part of the code the developer is currently working on and allows the agent to understand what areas of the code it should focus on to apply the manipulations (for example, changing code that is "close" to the cursor might affect the code assistant's suggestions more than manipulations in other areas).

The agent processes these states sequentially, applying manipulations in a loop over the segments. This segmentation strategy allows the agent to navigate and refine extensive prompts effectively, ensuring comprehensive coverage and strategic manipulations. More formally, let $p_t$ be a prompt after $t$ manipulations, which is divided into $n$ segments: $p_{t,0}, ..., p_{t,n-1}$. We calculate the state representation for the $p_t$ ($\sigma_t$) as follows: $i = t \bmod n$ is the location of the current segment to be manipulated at time $t$. Given $p_{t,i}$, the segment in the $i$th location at time $t$, the state embedding is: $\sigma_{p_{t,i}}^{emb} = \mathcal{L}(p_{t,i})$, where $\mathcal{L}(\cdot)$ is the pretrained encoder producing the embedding. We represent the segment's relative location as $\sigma_{p_{t,i}}^{loc}$ and the cursor relative position for $p_t$ as $\sigma_t^{cur}$.

The complete state representation $\sigma_t$ is the concatenation:

$$\sigma_t = [\sigma_{p_{t,i}}^{emb}, \sigma_{p_{t,i}}^{loc}, \sigma_t^{cur}] \tag{1}$$

**Observation Normalization.** Given that we used the last hidden states of pretrained LLM as observation representation, there might be slight differences across various samples. In order to deal with this, we keep a running calculation of the mean and standard deviation for these observations and normalize them prior to inputting them into the policy and value networks. Such normalization is a standard practice in RL, and we observed that it enhances the efficacy of our approach.

### 3.2.2 ACTIONS

We defined a set of manipulations (referred to as $\mathcal{A}$) designed to provide our agent with the flexibility to effectively manipulate the prompts. In each step, our agent can apply a manipulation on the prompt or decide to stop the prompt manipulation process and send it to the code assistant service without further changes. The full list of the agent's possible actions can be found in Appendix C.

### 3.2.3 REWARDS

The reward function $R_t$ is designed to capture the trade-off between minimizing code leakage and preserving suggestion relevance. During training, the agent interacts with the code assistant service, sending manipulated prompts and receiving suggestions. This allows computation of the reward $R_t$ based on the similarity between the original and manipulated suggestions, guiding the agent towards an optimal prompt manipulation policy. Note that the reward considers the similarity of a full prompt and the corresponding suggestion received for the prompt.

The reward function for training CodeCloak is based on the CodeBLEU metric Ren et al. (2020), which effectively captures the syntactic, structural, and semantic similarities between code segments. CodeBLEU extends the traditional BLEUPapineni et al. (2002) metric by incorporating domain-specific features of code, making it particularly suited for evaluating code generation and manipulation tasks. This metric score ranges are from 0% to 100%, where a higher score reflects greater similarity in terms of both code structure and functionality.

Let $p_t$ and $p_t'$ denote the original and manipulated prompts at timestep $t$, respectively. Let $s_t$ and $s_t'$ denote the suggestions generated by the code assistant model for $p_t$ and $p_t'$. The total reward score is defined as:

$$RScore(p_t, p_t', s_t, s_t') = \lambda_1 \cdot CodeBLEU(s_t, s_t') - \lambda_2 \cdot CodeBLEU(p_t, p_t') \tag{2}$$

$\lambda_1 > 0$ and $\lambda_2 > 0$ are hyperparams for the trade-off between the prompts' and the suggestions' similarity. We chose to use complementary values to balance the two components (i.e., $\lambda_2 = 1 - \lambda_1$). Naturally, the $RScore$ awards a positive reward when the similarity between the original and manipulated suggestions exceeds that of the prompts. Conversely, it awards a negative reward when the similarity between the suggestions is smaller than the similarity between the prompts. The goal is to optimize the $RScore$ for the final manipulated prompt.

In traditional RL, the objective is to optimize the cumulative reward $G_t$ over the entire MDP. However, in our prompt manipulation scenario, we are primarily interested in the performance of the

final manipulated prompt. Therefore, instead of using the $RScore$ directly, we propose using the difference in the $RScore$ between successive manipulations as the immediate reward:

$$R_t = RScore(p_t, p_{t'}, s_t, s_{t'}) - RScore(p_{t-1}, p_{t'-1}, s_{t-1}, s_{t'-1}) \tag{3}$$

To better reward high performance, we apply a scaling mechanism that uses multipliers based on predefined similarity ranges, boosting the total reward proportionally for more accurate results.

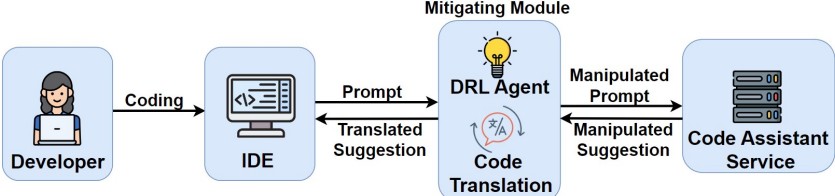

Figure 2: CodeCloak interactions.

**Reward Normalization.** To maintain consistency in the reward range, we apply an exponential moving average. We observed that this enhances the efficacy of our approach.

**Reward Shaping.** To enhance our agent's effectiveness, we implemented reward shaping by defining score ranges with scaling factors, which enable the agent to better balance privacy and relevance.

### 3.2.4 PROMPT MANIPULATION PROCESS

Given an initial prompt $p$, the agent starts in the state representation $\sigma_0$ as described above. In each timestep $t$, the agent selects an action $a_t$ from the set of possible manipulations $\mathcal{A}$, such as deleting or inserting lines, renaming variables or functions, or modifying functions. After applying the action $a_t$ to the prompt $p_t$, the agent transitions to a new state $\sigma_{t+1}$ representing the manipulated prompt $p_{t+1}$. Note that in each step, the agent manipulates one segment of the total prompt. However, a manipulation applied to one segment might influence other segments. For example, if a variable name is changed in one segment, the agent will replace that variable name in all other segments where that variable instance appears.

During the training process, The DRL agent interact with the code assistant model and receive the code suggestions by sending the manipulated prompts to the code assistant model. This allows the agent to measure the similarity between the suggestions received for the original and manipulated prompts, and calculate the reward required by the learning algorithm to train the DRL agent. The output of this phase is a trained DRL agent capable of effectively manipulating code prompts sent to an LLM-based code assistant in a manner in which the privacy of the code owner and the effectiveness/relevance of the code assistant's suggestions are preserved. Note that the training of the DRL agent is performed according to a policy which defines the preferred balance between preserving privacy (preventing the leakage of sensitive code) and preserving the relevance of suggestions made by the code assistant.

### 3.3 CODECLOAK TRAINING PHASE

In order to produce an efficient agent, capable of manipulating the prompts in an optimal manner, a DRL algorithm is used to train the agent (see Figure 2). The input to the DRL agent is a dataset of prompts from various projects, all generated by the code assistant client. During its training, the agent processes the prompts and iteratively selects an action to apply on the prompt until the 'stop manipulation' action is selected; eventually, this will enable the trained DRL agent to successfully manipulate unseen or complex prompts during the inference phase. To assess the effectiveness of the DRL agent's manipulations during the training process, the agent interacts with the code assistant model and receives the code suggestions by sending the manipulated prompts to the code assistant model. This allows the agent to measure the similarity between the suggestions received for the original and manipulated prompts, and calculate the *Reward* required by the learning algorithm to train the DRL agent. The output of this phase is a trained DRL agent capable of effectively manipulating code prompts sent to an LLM-based code assistant in a manner in which the privacy of the code owner and the effectiveness/relevance of the code assistant's suggestions are preserved. Note that the training of the DRL agent is performed according to a policy; this policy defines the preferred balance between preserving privacy (preventing the leakage of sensitive code) and preserving the original code suggestions made by the code assistant.

## 3.4 CodeCloak's Inference Phase

Following the training phase, the Mitigating component, which includes the DRL agent, is placed between the IDE and the code assistant service (see Figure 2). During coding, each prompt sent from the developer's IDE to the code assistant service is processed by the DRL agent before it is sent. The agent manipulates the prompt, minimizing the leakage while preserving the suggestion's relevance, and sends the manipulated prompts to the code assistant service. In some cases, an additional component called the Code Translation component, which is part of the Mitigating module, is applied to the returning suggestions of the code assistant service before it is sent back to the developer's IDE; for example, in cases in which the DRL agent changes the names of functions/variables in the prompts before sending them to the code assistant service, the Code Translation component maps the names of the functions/variables to the original ones so they will be aligned with the developer's code. The complete process of the CodeCloak agent for a given prompt in the inference stage is outlined in Algorithm 3.

In our research, we used recurrent proximal policy optimization (recurrent PPO) Schulman et al. (2017) as our RL model. We chose to use this model due to its ability to handle temporal dependencies effectively. This is crucial in sequential tasks, as prompt manipulations build on prior steps. Additionally, recurrent PPO enables us to feed different segments of the source code to the model at each step within an episode, allowing the agent to make decisions based on evolving context throughout the episode.

Figure 3: CodeCloak Inference Stage.

**Input:** original prompt $p$, CodeCloak policy $\pi_{\text{CodeCloak}}$
**Output:** CodeCloak's suggestion $s'$
$p_0 \leftarrow p$
Divide prompt $p_0$ into $n$ segments $p_{0,0}, \ldots, p_{0,n-1}$
$t \leftarrow 0$
**while** True **do**
    $i \leftarrow t \bmod n$
    $\sigma_t \leftarrow$ state representation for $p_{t,i}$
    $a_t \leftarrow \pi_{\text{CodeCloak}}(\sigma_t)$
    **if** $a_t =$ "StopManipulations" **then**
        **break**
    **else**
        $p_{t+1,i} \leftarrow$ apply $a_t$ on $p_{t,i}$
        $t \leftarrow t + 1$
$s' \leftarrow$ CodeAssistant suggestion for $p_t = p_{t,0}, \ldots, p_{t,n-1}$
$s' \leftarrow$ apply CodeTranslation on $s'$
**return** $s'$

## 4 Evaluation

### 4.1 Evaluation Setup

**Models.** In our experiments, we used local versions of the following LLM-based code assistant models: StarCoder Li et al. (2023) and Code Llama-13B Roziere et al. (2023). Due to GitHub Copilot's policy prohibiting automated processes that manipulate and analyze the model's behavior, we were unable to use CodeCloak to manipulate prompts sent to Copilot. Therefore, we trained CodeCloak using StarCoder and evaluated it using the StarCoder and Code Llama models.

**Datasets.** To train and evaluate CodeCloak, we created a dataset by randomly sampling repositories in the Python language from the CodeSearchNet Husain et al. (2019) dataset. For each repository, we simulated a coding process by navigating to random locations in the repository, copying a random number of code lines to the IDE, and using our *Developer Coding Simulator* which are explained in Appendix E, simulating typing from this point for a short period of time (between 20-90 seconds), capturing the generated prompts. This process yielded a dataset of around 30K diverse prompts of various sizes from around 500 repositories, allowing CodeCloak to learn how to manipulate prompts effectively. To assess the effectiveness of CodeCloak, we used prompts containing more than four functions, since they contain a substantial amount of code that can be leaked and allow the agent to apply manipulations.

**CodeCloak Details.** To train the DRL agent, we used the Recurrent PPO algorithm from the Stable Baselines3 Contrib library Raffin et al. (2021). The state embedding was obtained by using the last hidden state of the StarEncoder[5] model. The maximum token limit for input prompts was set at 1,600 tokens. To accommodate prompts that exceed the StarEncoder model's maximum token limit, we divided each prompt into two segments. The output response length was set at 20 tokens. In addition, unless otherwise specified, the hyperparameters $\lambda_1$ and $\lambda_2$ were both set at 0.5, giving equal importance to both of CodeCloak's objectives.

---

[5] https://huggingface.co/bigcode/starencoder

## 4.2 EVALUATION METRICS

**CodeBLEU.** The CodeBLEU Ren et al. (2020) score extends the BLEUPapineni et al. (2002) as descried before. To evaluate CodeCloak using CodeBLEU, we set the weights of CodeBLEU components as proposed in the original paper Ren et al. (2020) to be the most suitable for measuring similarity in code refinement tasks, capturing the preservation of the code's semantic meaning and functional behavior: $\alpha, \beta, \gamma, \delta = 0.1, 0.4, 0.1, 0.4$.

**LLM as a Judge.** LLMs have been found to be effective evaluators for a wide range of tasks, aligning well with human preferences Zheng et al. (2023). In particular, Zheng et al. reported that GPT-4 achieves over 80% agreement with human judgments, establishing it as a reliable tool for evaluation. Building on this, recent studies have explored the use of LLMs, specifically ChatGPT, in various code-related applications, showcasing its potential for code analysis. For instance, methods using ChatGPT for automatic code documentation Khan & Uddin (2022), code summarization and refinement Guo et al. (2024); Sun et al. (2023) tasks have demonstrated its ability to understand code semantics. ChatGPT's successful use in static code analysis and application security testing Bakhshandeh et al. (2023); Sadik et al. (2023), highlight its code "understanding" capabilities. Leveraging ChatGPT's strengths in code-related tasks, we used ChatGPT 4 (1106-preview version) to measure the similarity between code segments and assess CodeCloak's manipulation effectiveness through two metrics; for each metric we created a dedicated prompt: *PrivacyGPT* measures the percentage of the original code that was successfully protected from leakage, ranging from 0% (fully leaked) to 100% (no leakage), based on the comparison between the original code segment and the version obtained by an adversary. A higher score indicates that a larger portion of the original code was effectively prevented from being exposed. *SugSimGPT* - Given the original prompt and two code suggestions (the original suggestion and the one received for the manipulated prompt), ChatGPT provides a similarity score between 0% and 100%. A higher score indicates greater similarity between the suggestions. For each prompt, we instructed ChatGPT to consider different factors such as functional similarity, visual similarity, and the intended purpose or code meaning.

## 4.3 CODECLOAK'S RESULTS

**Our Agent's Effectiveness.** To evaluate CodeCloak's effectiveness, we compare its performance against two baseline models: (1) a *Random baseline*, where manipulations are applied randomly for each prompt, with the number of actions ranging up to the maximum possible numuber of manipulations, and (2) an *LLM-based baseline*, which leverages a large language model (specifically ChatGPT-4) to dynamically determine both the type and extent of manipulations. In the LLM-based baseline, ChatGPT-4 is provided with a system prompt explaining the objective of reducing code leakage while maintaining suggestion relevance, allowing it to make informed, context-aware decisions. The results, as shown in Table 1, demonstrate CodeCloak's effectiveness in balancing code leakage mitigation and preserving suggestion relevance. For the CodeBLEU metric on suggestions, CodeCloak achieved a significantly higher score of 72.3%, compared to 49.2% for the random baseline and 42.6% for the LLM-based baseline, demonstrating its superior ability to maintain suggestion quality. Additionally, CodeCloak excelled with an 80.2% score for the SugSimGPT metric, outperforming both the random baseline (46.2%) and the LLM-based baseline (38.7%), further indicating the high relevance of the suggestions generated after prompt manipulations.

While the random baseline performed slightly better in terms of code leakage, this improvement came at the expense of suggestion quality, where CodeCloak clearly excels. Similarly, while the LLM-based baseline achieved a balanced trade-off between leakage mitigation and suggestion relevance, its performance was still inferior to that of CodeCloak, which optimizes this balance more effectively. An analysis of the actions taken by CodeCloak reveals that it tends to apply more aggressive manipulations, such as 'Delete Functions' Body,' during the initial steps of the process. This indicates that the agent prioritizes reducing leakage by focusing on critical aspects of the prompt while still preserving the utility of the suggestions.

An analysis of the actions taken by CodeCloak reveals that it tends to apply more aggressive manipulations, such as 'Delete Functions' Body' during the initial steps of the process. This indicates that the agent prioritizes reducing leakage by focusing on significant modifications early on. As the process progresses, the agent transitions to more refined actions, such as 'Change Variables Names', which fine-tune the suggestions while preserving relevance. A heatmap illustrating the action selection distribution can be found in Appendix G.

**Prioritizing Suggestions and Requests.** In addition to the baseline model with equal prioritization ($\lambda_1 = \lambda_2 = 0.5$), we trained two additional models to explore the trade-offs between mitigating

Table 1: Performance comparison of the random and CodeCloak approaches

| | CodeBLEU | | ChatGPT Metrics | |
| --- | --- | --- | --- | --- |
| | Prompts ↓ | Suggestions ↑ | PrivacyGPT ↑ | SugSimGPT ↑ |
| **Random baseline** | **35**% | 49.2% | **62.4**% | 46.2% |
| **LLMs based baseline** | 41.1% | 42.6% | 53.9% | 38.7% |
| **CodeCloak** | 36.4% | **72.3%** | 58.2% | **80.2%** |

Table 2: CodeCloak's performance for different prioritizations

| | CodeBLEU | | ChatGPT Metrics | |
| --- | --- | --- | --- | --- |
| | Prompts ↓ | Suggestions ↑ | PrivacyGPT ↑ | SugSimGPT ↑ |
| Equal prioritization | 36.4% | 72.3% | 58.2% | 80.2% |
| Prioritizing Suggestions | 68% | 82.5% | 24.9% | 90.7% |
| Prioritizing Requests | 30.1% | 50.2% | 76.2% | 55.3% |

leakage and preserving functionality: (1) *Prioritizing Suggestions*: A model that prioritizes the relevance of suggestions over mitigating leakage, configured with $\lambda_1 = 0.3$ and $\lambda_2 = 0.7$. This setup ensures developers receive high-quality, useful suggestions despite prompt manipulations, focusing on enhancing coding efficiency. (2) *Prioritizing Requests*: A model that emphasizes mitigating code leakage over preserving the relevance of suggestions, configured with $\lambda_1 = 0.7$ and $\lambda_2 = 0.3$. This setup ensures that manipulated prompts effectively hide and protect the developer's original code. These two configurations allowed us to investigate the impact of different priorities on the effectiveness of our approach in real-time coding environments.

Table 3: CodeCloak's performance of in a cross-LLM transferability setup

| | CodeBLEU | | ChatGPT Metrics | |
| --- | --- | --- | --- | --- |
| | Prompts ↓ | Suggestions ↑ | PrivacyGPT ↑ | SugSimGPT ↑ |
| **StarCoder** | 36.4% | 72.3% | 58.2% | 80.2% |
| **CodeLlma 13B** | 36.2% | 73.8% | 58.2% | 78.3% |

The results presented in Table 2 demonstrate how CodeCloak can be adjusted for different component weights, depending on the developer's needs.

**Transferability.** As can be seen in the results presented in Table 3, CodeCloak demonstrates full transferability between StarCoder and Code Llama (i.e., training the DRL agent on StartCoder and using the trained agent on Code Llama), which emphasizes its robustness and usefulness.

### 4.4 EVALUATING CODECLOAK'S SUGGESTIONS USING HUMAN JUDGMENT

To further emphasize the relevance of the suggestions provided by CodeCloak, we conducted a user study involving human judgment. We created two separate questionnaires, each containing 20 prompts, along with two corresponding suggestions for each prompt. In Questionnaire 1, participants were shown code suggestions from StarCoder for both the original prompt and the prompt manipulated by CodeCloak. Questionnaire 2 followed the same format, but instead of presenting the suggestions that were generated by CodeCloak, we present the suggestion created by baseline model (see 4.3). Participants were tasked with evaluating the relevance of the suggestions without knowing which was original or manipulated. For each prompt, developers were shown both suggestions side by side (in random order) and were asked to indicate which suggestion they found more relevant, using a five-point scale where one side represented preference for the first suggestion and the other for the second. After collecting their responses, we analyzed the results by mapping their ratings to a scale from -2 to 2, where -2 indicates that the suggestion for the original prompt is more relevant, 0 indicated equal relevance and 2 indicates that the suggestion for the manipulated prompt is more relevant. Then, for each prompt, we averaged the developers' rankings to produce a final relevance score. We distributed the questionnaires to 20 experienced developers, with 10 participants evaluating each set of 20 prompts. The prompts used in this evaluation were randomly selected from the test set of the evaluation dataset. The results from the questionnaires closely align with the previous metrics, reinforcing CodeCloak's ability to preserve suggestion quality. In Questionnaire 1, 13 out of 20 prompts (65%) rated the suggestions from the manipulated prompts as equal to or better than the original ones, with a mean score of -0.24 and a standard deviation of 0.54, indicating only a slight preference for the original suggestions. In contrast, Questionnaire 2 demonstrated that only 4 out of 20 prompts (20%) rated the suggestions from the baseline manipulations as equal or

better, with a significantly lower mean score of -1.02 and a standard deviation of 0.72. These results clearly show that CodeCloak produces significantly more relevant and useful suggestions than those created by the baseline model, while effectively reducing code leakage.

## 4.5 EVALUATING CODE LEAKAGE ON COMPLETE REPOSITORIES

**Code Reconstruction.** To assess CodeCloak's effectiveness in reducing code leakage, we employed a dynamic code reconstruction method that simulates how an attacker could potentially reconstruct a developer's complete repository from intercepted prompts during the coding process. The method includes the following stages: *(1) Prompt Interception and Monitoring:* During the developing simulated process, we capture all prompts sent from the IDE to the code assistant over time. *(2) Aggregating and Reconstructing the Codebase:* Using an LLM model with a detailed reconstruction instructions prompt, we dynamically aggregated the captured prompts to reconstruct the evolving codebase, reflecting changes and modifications made during development. This approach accounts for changes and modifications made during development. *(3) Comparative Analysis:* We compared the reconstructed code with and without CodeCloak to measure code leakage mitigation.
Prompts were intercepted using BURP Suite[6] within the PyCharm IDE. To reconstruct a codebase from the code segments, we used ChatGPT 4.0 (1106-preview version). Additional details regarding this experiment can be found in Appendix D.

**Evaluating CodeCloak on Complete Repositories** To evaluate CodeCloak 's real-world effectiveness, we conducted experiments on five randomly selected GitHub repositories of varying sizes, created after the formal cutoff date for ChatGPT-4 training to minimize the likelihood of overlap with its training data. Each repository contained 3-4 files, ranging from 30 to 900 lines (see Appendix F). For each repository, we simulated the coding process twice: once without CodeCloak and once with it. Due to the sequential nature of prompts during development, we applied a pre-processing step to ensure consistent manipulations across the codebase. Without CodeCloak, the reconstructed repositories achieved an average CodeBLEU score of 79.1%, indicating substantial code leakage. However, with CodeCloak applied, the average CodeBLEU score for these repositories dropped significantly to 40.1%, highlighting a notable reduction in code exposure. This preprocessing step identified previously manipulated code segments and applied the same changes to subsequent prompts. After preprocessing, CodeCloak further manipulated the prompts to minimize code leakage while maintaining the relevance and usefulness of suggestions. Finally, We then employed our *code reconstruction method* to evaluate the extent of code leakage, reconstructing the codebase using the captured prompts with and without CodeCloak. Additionally, the *SugSimGPT* metric, which leverages ChatGPT's code analysis capabilities, produced an average similarity score of 74.3%, showing that the relevance of the suggestions remained largely intact. The CodeBLEU

Table 4: CodeCloak's performance on complete repositories

| | Avg CodeBLEU Complete Repos↓ | CodeBLEU Suggestions ↑ | SugSimGPT ↑ |
|---|---|---|---|
| **Without CodeCloak** | 79.1% | **100%** | **100%** |
| **With CodeCloak** | **40.1%** | 68.4% | 74.3% |

score for suggestions averaged 68.4%, further confirming CodeCloak 's effectiveness in preserving the usefulness of the code assistant's suggestions during the development process.

## 5 CONCLUSION

This study addressed the critical issue of code leakage risk resulting from the widespread adoption of LLM-based code assistants. We proposed CodeCloak, a novel DRL agent that manipulates prompts sent to code assistant services to mitigate this risk. CodeCloak effectively reduces code leakage while preserving the relevance and the usefulness of the code assistant's suggestions. These contributions pave the way for more secure and privacy-preserving use of LLM-based code assistants. By offering a practical solution for mitigating leakage during the development process, CodeCloak facilitates the responsible adoption of these powerful tools. We anticipate that our work will serve as a catalyst for further research into the security and privacy challenges associated with LLM-based code assistants.

---

[6]https://portswigger.net/burp/documentation/desktop/tools/proxy

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

# A  APPENDIX

# B  DEEP REINFORCEMENT LEARNING

Reinforcement learning (RL) is an area of machine learning (ML) that addresses multi-step decision-making processes. An RL algorithm normally consists of an agent that interacts with an environment in a sequence of actions and rewards. In each time step $t$, the agent selects an action $a_i \in A = \{a_1, a_2, ..., a_k\}$. As a result of $a_i$, the agent transitions from the current state $s_t$ to a new state $s_{t+1}$. The selection of the action may result in a reward, which can be either positive or negative. The goal of the agent in each state is to interact with the environment in a way that maximizes the sum of future rewards $G_t$. Eq. 4 represents the cumulative reward over time:

$$G_t = \sum_{k=0}^{\infty} \gamma^k R_{t+k+1} \tag{4}$$

where:

- $G_t$ is the expected return at step $t$.
- $R_{t+k+1}$ represents the reward received after $k$ steps from step $t$.
- $\gamma \in [0, 1]$ is the discount factor, which provides the present value to future rewards. A reward received $k$ steps in the future is valued at $\gamma^k$ times its immediate value.

In order to maximize Eq. 4, the agent needs to learn its optimal policy. The policy defines how the agent behaves at any given time; i.e., which action to select when in a given state. When faced with complex environments with high-dimensional data, it is common to use DRL, i.e., incorporate deep neural networks to learn the optimal policy. This enables the agent to learn more sophisticated strategies and make more informed decisions. DRL algorithms can efficiently explore vast state and action spaces and devise strategies for addressing complex problems. Moreover, DRL excels in handling partial and noisy information. Utilizing the reward function, DRL algorithms can reconcile multiple, sometimes conflicting objectives, making them invaluable in security-related scenarios.

# C CODECLOAK ACTION LIST

Table5 contains a list of all the possible actions that CodeCloak can apply in each step.

Table 5: CodeCloak's code manipulation actions

| Action | Description |
|---|---|
| Detect and Replace PII | Detect PII (e.g., names, API keys, emails) in the current code segment and replace them with random strings of a fixed length. |
| Change Random Lines | Swap positions of two random lines present in the current code segment. |
| Delete Random Line | Remove a random line from the current code segment. |
| Insert Random Line | Insert a random line in the current code segment. |
| Delete Functions' Body Incrementally | Incrementally remove function bodies from the current code segment and replace the removed body replaced with summarization comments (that were generated using a local summarization model). |
| Delete Functions' Body (Keep Last) | Remove all function bodies except the last one in the current code segment and replace the removed bodies with summarization comments (that were generated using a local summarization model). |
| Delete Functions' Body | Remove all function bodies present in the current code segment and replace them with summarization comments describing the original function bodies (that were generated using a local summarization model). |
| Delete Functions Incrementally | Incrementally remove functions from the current code segment. |
| Change Functions Names | Replace all function identifiers in the current code segment with random strings of a fixed length (5 characters) consisting of letters and numbers. |
| Change variables names | Replace all variable identifiers in the current code segment with random strings of a fixed length (5 characters) consisting of letters and numbers. |
| Change Arguments Names | Replace all argument identifiers in the current code segment with random strings of a fixed length (5 characters) consisting of letters and numbers. |
| Stop Manipulations | Stop processing the current prompt and send it to the code assistant service without further manipulations. |

## D CODE RECONSTRUCTION METHOD

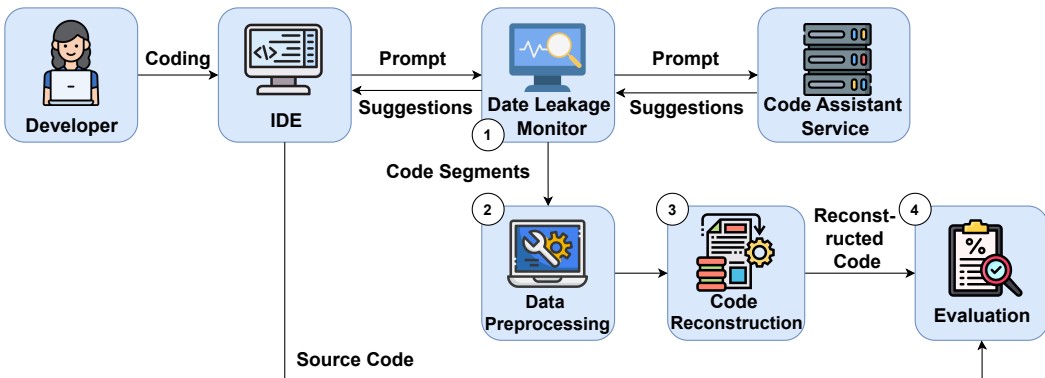

Figure 4: An illustration of the proposed code reconstruction process long with its main components.

### D.1 PROBLEM DEFINITION

In this section, we introduce our method for code reconstruction using prompts when using LLM-based code assistants. Our method aims to evaluate the extent of codebase leakage during the development process. In order to achieve this, three challenges must be addressed:

1. The need to determine an effective method for intercepting and monitoring prompts sent to the code assistant service during the development process;

2. The need to formulate and design a code reconstruction approach capable of reconstructing the developer's codebase, based on LLMs, in which the prompts are aggregated over time; and

3. The need to define a method for evaluating the severity of the leakage, given the original and reconstructed codebases.

The main challenge in reconstructing code from the prompts sent to the code assistant over time stems from the frequent modifications made by the developer during coding, such as editing and deleting code segments in different locations. Because of such modifications, early prompts that have been leaked and captured may be irrelevant and outdated or duplicated, preventing the use of traditional reconstruction methods that combine all available code segments into one complete codebase Schauer et al. (2010); Shanmugasundaram & Memon (2002).

The main components of our proposed method are presented in Figure 4.

### D.2 DATA COLLECTION AND PREPARATION

To collect the prompts during the development process, the Data Leakage Monitor component (component 1 in Figure 4) intercepts the traffic between the IDE and the code assistant service and extracts the prompts that are being sent from the code assistant client to the service. This process is performed in real time, for example, by placing a proxy (such as Burp Suite Junmei & Chengkang (2021)) between the IDE and the code assistant service.

In the next step, the proposed method performs a preparatory process, using the Data Preparation component (component 2 in Figure 4), which is focused on analyzing the prompts. Its goal is to extract the code segments and additional contextual data (for example, in some code assistants, the prompt includes the source files' names from which the code segments were extracted). In cases where the prompt includes this information, we process the prompts by aggregating them by the file they belong to (based on the filename).Next, we order the prompts chronologically (by their sending timestamp). At this stage, for each file, we have an ordered sequence of code segments. Finally, we remove code segments that are contained within subsequent code segments to avoid duplication and reduce code reconstruction time. By removing redundant code segments, we streamline the reconstruction process, enabling out method to receive a more comprehensive and concise representation of the codebase segments, resulting in a more accurate code reconstruction.

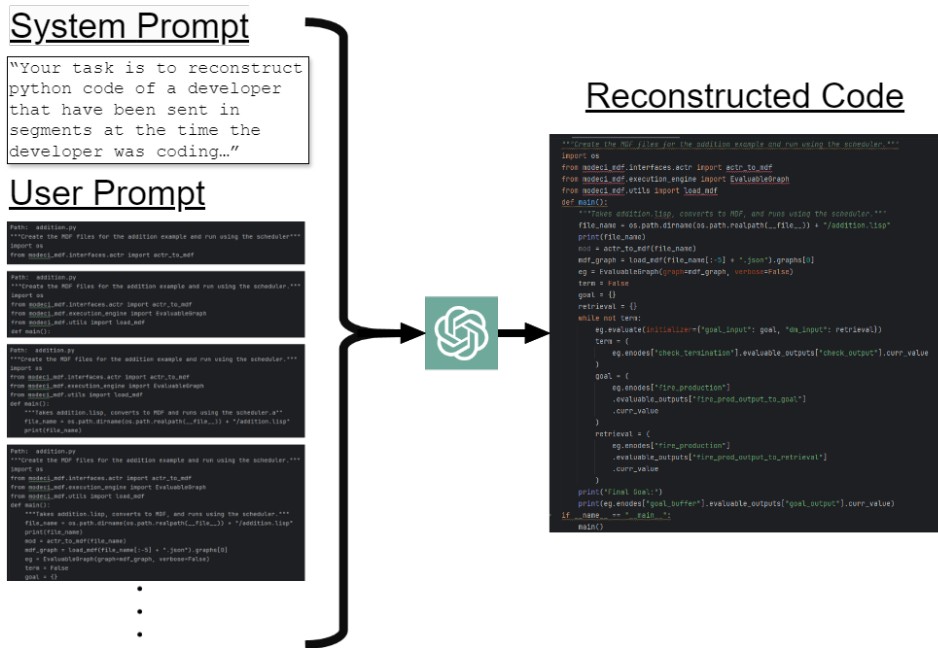

Figure 5: System and user prompts for code reconstruction using ChatGPT.

## D.3 CODE RECONSTRUCTION

Following the data collection and preparation phase, our method employs an LLM, which serves as the Code Reconstruction component (component 3 in Figure 4). As input, this model receives the processed prompts in the order in which they were transmitted from the code assistant client (in the IDE) to the code assistant service. In case where the total segments' size exceeds the max token limit the LLM is able to process, we send the LLM segments of each file separately. If the total segments' size still exceeds the max token limit, we perform the following iterative process: we take the maximum possible segments (that do not exceed the max token limit) and send them to the LLM. In the next iteration, we provide the LLM the codebase reconstructed in the last step, along with new segments. We repeats this process until we complete processing all the segments.The Code Reconstruction component then combines these code segments, building a representation of the developer's final code. When these prompts are submitted to the LLM (ChatGPT4 in our evaluation), they are accompanied detailed instructions that clearly define the task (see Figure 5). This is done to ensure that the model produces relevant and accurate output consisting of the name and content of each file identified, providing a comprehensive view of the leaked code.

## D.4 LEAKAGE EVALUATION

After reconstructing the codebase from the collected code segments, our next step is to evaluate the effectiveness of the code reconstruction process and the amount of code leakage (component 4 in Figure 4). This involves assessing whether the proposed method accurately reconstructed the code in its original form and whether the reconstructed version clearly conveys the original intent and functionality of the code. This evaluation process allows our method to accurately determine the percentage of the developer's code that was leaked. In our evaluation, we use the normalized edit distance metric, which measures how similar two texts are by calculating the least number of changes needed to transform one text into another, adjusted by their lengths. This offers a straightforward, scaled gauge for visually comparing text differences and clearly depicting variations or mistakes.

## E   DEVELOPER CODING SIMULATOR

We developed a Python-based simulator designed to mimic the realistic behavior of developers during the coding process. The simulator incorporates features such as introducing coding errors at random intervals, correcting those errors after writing additional lines of code, simulating pauses in typing, going back and forth in the file, and maintaining a developers' average typing speed rate. This simulator was used to generate datasets to train and evaluate CodeCloak. It operates within an IDE configured with code assistant plugins, allowing us to capture the prompts sent from the IDE to the code assistant servers and the corresponding suggestions received. The prompts and suggestions captured during these simulated coding sessions formed the datasets used to evaluate our mitigation method across different code assistants, including StarCoder and Code Llama models.

## F   GITHUB DATASET

In Table 6 we list the repositories used in our experiments (we randomly selected 5 repositories with the condition of 3-4 files in each repository).

Table 6: GitHub repositories used in our experiments

| # | Github File Path |
|---|---|
| 1 | `https://github.com/robert-koch-institut/mex-backend/blob/main/tests/test_roundtrip.py`
`https://github.com/robert-koch-institut/mex-backend/blob/main/tests/test_exceptions.py`
`https://github.com/robert-koch-institut/mex-backend/blob/main/tests/identity/test_provider.py`
`https://github.com/robert-koch-institut/mex-backend/blob/main/tests/graph/test_query.py` |
| 2 | `https://github.com/stocknear/backend/blob/main/app/twitter.py`
`https://github.com/stocknear/backend/blob/main/app/ta_signal.py`
`https://github.com/stocknear/backend/blob/main/app/rating.py`
`https://github.com/stocknear/backend/blob/main/app/market_movers.py` |
| 3 | `https://github.com/zinabkaviani/Trellomize/blob/main/Codes/globals.py`
`https://github.com/zinabkaviani/Trellomize/blob/main/Codes/register.py`
`https://github.com/zinabkaviani/Trellomize/blob/main/Codes/User/user.py` |
| 4 | `https://github.com/ScanMountGoat/xenoblade_blender/blob/main/xenoblade_blender/import_camdo.py`
`https://github.com/ScanMountGoat/xenoblade_blender/blob/main/xenoblade_blender/__init__.py`
`https://github.com/ScanMountGoat/xenoblade_blender/blob/main/xenoblade_blender/export_root.py`
`https://github.com/ScanMountGoat/xenoblade_blender/blob/main/xenoblade_blender/export_wimdo.py` |
| 5 | `https://github.com/ToTheBeginning/PuLID/blob/main/eva_clip/loss.py`
`https://github.com/ToTheBeginning/PuLID/blob/main/eva_clip/hf_model.py`
`https://github.com/ToTheBeginning/PuLID/blob/main/eva_clip/modified_resnet.py`
`https://github.com/ToTheBeginning/PuLID/blob/main/eva_clip/timm_model.py` |

## G   CODECLOAK ACTION DISTRIBUTION HEATMAP

The heat map in Figure 6 presents the distribution of different manipulations/actions ($x$-axis) chosen by CodeCloak across various time steps (y-axis) during the evaluation process. Note that the heatmap refers to sequences of manipulations of variable length.

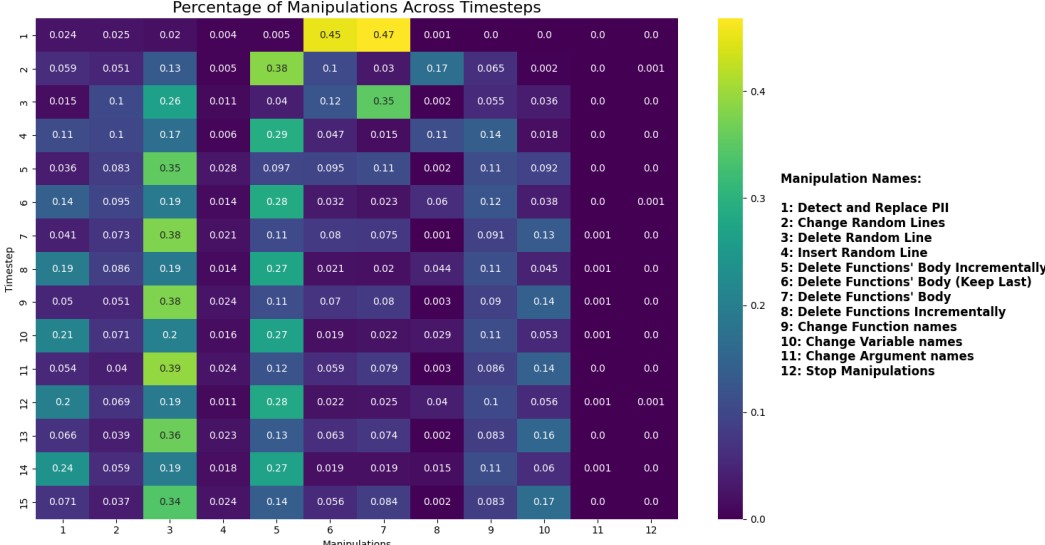

Figure 6: CodeCloak Action Distribution Heatmap.

# H HYPERPARMAMETERS FOR TRAINING CODECLOAK

The table below provides the training details for CodeCloak. In addition, We also adopt parallel environment in our training process (eight environments). We have noticed that large size of parallel environments helped for our training process.

| Hyperparameter | Value |
| --- | --- |
| Learning Rate | 0.00025 |
| Time Limit (per episode) | 15 |
| Gamma (Discount Factor) | 0.99 |
| GAE Lambda | 0.95 |
| N Steps | 128 |
| Batch Size | 64 |
| Clip Range | 0.2 |
| Entropy Coefficient | 0.01 |
| Policy Net Arch | Policy Network: [256, 256, 256, 128], Value Network: [256, 256, 256, 128] |

Table 7: Hyperparameters and Policy Configuration for Training CodeCloak

