# OpenReview forum: "CodeCloak: A Method for Mitigating Code Leakage by LLM Code Assistants"
_ICLR.cc/2025/Conference — Submitted to ICLR 2025_

### Official Review · Reviewer_HNzE · 2024-10-25

**Soundness:** 2
**Presentation:** 2
**Contribution:** 2
**Rating:** 5
**Confidence:** 3

**Summary:**

This paper proposed CodeCloak, a deep reinforcement learning-based technique to manipulate code prompts to mitigate the risk of code exposure. In particular, CodeCloak consists of states, actions and rewards, which are common in RL. For states, a code encoder embeds the prompt, and the position information is also included in the constructed embeddings. For actions, it defines several code manipulation actions including delete, change. For rewards, the reward function is based on the CodeBLEU metric. Some experiments are conducted to evaluate the effectiveness of the proposed technique.

**Strengths:**

- The security problem is important and meaningful in the era of LLMs.
- Easy to follow.

**Weaknesses:**

- The technique contributions are weak. The used RL is standard, which includes actions and rewards.
- The evaluation is weak and misses state-of-the-art baselines to confirm the effectiveness of the proposed techniques.
- The motivation for using RL should be strengthened.
- Code and data are not open-sourced.

**Questions:**

- Why use RL in this work? What are the advantages of using RL to avoid data leakage?
- How to define the actions in code completion, any missing actions in the defined sets?
- What choose StarEncoder for prompt embedding? Any experiments to compare its performance with other encoders like CodeBERT?

---

> ### Author Response · Authors · 2024-11-21
>
> Dear reviewer HNzE,
>
> Following your comment related to *baseline comparison,* we performed (and will add to the revised paper) an additional baseline using ChatGPT-4. In this baseline, we provided ChatGPT-4 with a system prompt detailing the objective of reducing code leakage while maintaining the quality of suggestions. The ChatGPT agent was unconstrained and could choose any manipulation it deemed suitable to apply on the prompt. We then evaluated the manipulated prompts generated by ChatGPT-4 and the corresponding responses against our method. The CodeBLEU metric for this approach resulted in 41.1% for the prompts, compared 36.4% in our method, (*lower is better*) to and 42.6% for the responses, compared to 72.3% in our method (*higher is better*).
>
> We thank the reviewer for this comment as we believe that this baseline (in addition to the random one) further demonstrates the advantage of our proposed DRL-based method.

---

### Official Review · Reviewer_NfvR · 2024-10-31

**Soundness:** 2
**Presentation:** 1
**Contribution:** 2
**Rating:** 5
**Confidence:** 3

**Summary:**

The paper presents CodeCloak, a reinforcement learning-based method for mitigating code leakage in LLM-based code assistants. The approach manipulates prompts by applying various transformations to preserve the confidentiality of the developer's proprietary code while maintaining the quality of the code suggestions. Using a DRL agent, CodeCloak selectively manipulates prompts to minimize sensitive information exposure, tested across models like StarCoder and Code Llama. The evaluations focus on a coding simulator and metrics like CodeBLEU and two GPT-based similarity measures to validate the system’s effectiveness in balancing privacy and usability.

**Strengths:**

Major Strengths

1. Timely response to an important issue: The paper addresses the growing concern of code leakage through LLM-based code assistants, a pertinent issue given the increasing adoption of such tools in development. CodeCloak’s focus on prompt manipulation as a privacy-preserving mechanism is both timely and relevant.

2. Direct and practical solution: The authors define the problem of code leakage through LLM prompts clearly and systematically approach it with DRL-based prompt manipulation. The solution, CodeCloak, is a streamlined method that operates locally without altering the assistant model, making it more practical for real-world applications.

Minor Strengths

1. Clear figure presentation: The figures are effectively used, especially the workflow and action selection heatmap, which illustrate the core operations of CodeCloak and the logic behind its prompt manipulations. These visuals enhance the clarity of the methodology.

2. The design of developer simulator: The developer coding simulator is a thoughtful addition, simulating real-world coding behaviors like pauses, cursor movement, and typo corrections. This setup provides a controlled yet realistic environment for testing CodeCloak, making the results more relatable and grounded in practical usage​.

3. Multi-dimensional evaluation strategy: By employing CodeBLEU[1], GPT-based similarity measures, and user studies, the authors take a holistic approach to evaluation. This variety provides a solid foundation for assessing CodeCloak’s ability to mitigate leakage without compromising functionality, making the claims more convincing.

[1] Ren S, Guo D, Lu S, et al. Codebleu: a method for automatic evaluation of code synthesis[J]. arXiv preprint arXiv:2009.10297, 2020.

**Weaknesses:**

Major Weaknesses

1. Limited technical contribution: While CodeCloak addresses a relevant problem, its technical novelty is limited, as it primarily combines established RL methods (specifically recurrent PPO [1]) with prompt manipulation techniques. This reliance on recurrent PPO introduces challenges, such as difficulty in reward modeling and a complex training pipeline. Additionally, selecting a specific RL algorithm may restrict the generality of the approach. Expanding the RL architecture or incorporating more advanced prompt adaptation methods could enhance the contribution and broaden its applicability.

2. Over-reliance on similarity for effectiveness assessment: Relying solely on CodeBLEU for code similarity may not capture the full effectiveness of the method. CodeBLEU assesses syntactic and structural similarity, which could miss nuanced privacy risks. While PrivacyGPT and SugSimGPT provide insights into leakage and suggestion quality, PrivacyGPT may not fully evaluate the sensitivity of the leaked information, and SugSimGPT focuses on relevance rather than privacy-preserving quality. Adding targeted metrics for prompt leakage or sensitivity would provide a more complete evaluation of CodeCloak’s impact.

3. Inconsistent formula presentation: The mathematical expressions, especially around the reward function, could benefit from clearer notation and consistent formatting. Improving this aspect would make the technical approach more rigorous and easier to follow. For example:

- The cumulative reward formula $G_t = \sum_{k=0}^{\infty} \gamma^k R_{t+k+1}$ lacks definitions for variables like $G_t$ and $R_{t+k+1}$.

- The agent’s action distribution formula is shown in the action heatmap, but it doesn’t fully explain the state-action mapping, which is essential for understanding decision-making.

- In Table 2, parameters $\lambda_1$ and $\lambda_2$ are used to balance relevance and leakage but lack explanation on how each impacts the model’s outputs.

These can make the paper hard to follow and create confusion. Clearer notation and additional context would improve readability and technical rigor.

Minor Weaknesses

1. Repeated sentences and paragraphs: The paper contains multiple repetitions, particularly around the reward process, DRL modeling, and LLM choices (e.g., StarCoder[2] vs. Code Llama[3]). Streamlining these explanations could improve clarity and reduce redundancy.

2. Possible issues in updating the DRL module: The paper mentions using PPO for the DRL agent but does not provide sufficient detail on how the module adapts to diverse prompt types. Expanding on how the DRL module handles varied developer prompts and how it recalibrates under different conditions would make the approach more robust and credible.

[1] Pleines M, Pallasch M, Zimmer F, et al. Generalization, mayhems and limits in recurrent proximal policy optimization[J]. arXiv preprint arXiv:2205.11104, 2022.

[2] Li R, Allal L B, Zi Y, et al. Starcoder: may the source be with you![J]. arXiv preprint arXiv:2305.06161, 2023.

[3] Roziere B, Gehring J, Gloeckle F, et al. Code llama: Open foundation models for code[J]. arXiv preprint arXiv:2308.12950, 2023.

**Questions:**

My primary concern lies in the technical contribution of this work, particularly in the modeling approach. To clarify this aspect, I would appreciate if the authors could address the following questions:

1. Could the authors elaborate on the specific innovations introduced in the DRL modeling process for CodeCloak beyond existing reinforcement learning techniques? For instance, how does the architecture adapt to unique challenges in privacy preservation for code prompts?

2. How does CodeCloak’s reward function balance code similarity with privacy protection, and was any tuning process developed to optimize this balance? Detailed insights into parameter selection and adjustments would be helpful (i.e., if more than a linear combination, what do you plan to change)

3. Given the dependency on CodeBLEU and GPT-based metrics, did you consider alternative metrics tailored to privacy risk or sensitivity? If so, why were these not implemented, and how might future versions of CodeCloak address this gap?

**Details Of Ethics Concerns:**

There are no ethical concerns with this work.

---

### Official Review · Reviewer_XBNY · 2024-11-04

**Soundness:** 2
**Presentation:** 2
**Contribution:** 2
**Rating:** 3
**Confidence:** 3

**Summary:**

The paper presents "CodeCloak," a novel method leveraging deep reinforcement learning to minimize the exposure of proprietary code in the use of LLM-based code assistants, such as StarCoder and Code Llama. By manipulating code prompts before submission, the method aims to secure proprietary code from potential leaks while maintaining the utility of the code assistant's responses. The authors demonstrate CodeCloak's effectiveness across multiple LLM models and code repositories, achieving a significant reduction in code leakage with minimal loss of suggestion relevance.

**Strengths:**

1. The proposal of a deep reinforcement learning agent to address code leakage in LLM-based code assistants is both timely and innovative, aligning well with current industry needs.
2. The focus on mitigating real-world risks associated with proprietary code in commercial settings is highly relevant and adds considerable value to the research.

**Weaknesses:**

Major Comments:
1. The metrics used for evaluating the preservation of code intent are not adequately justified. The reliance solely on edit distance may not effectively capture the semantic preservation of the code, which is crucial for assessing leakage risks.
2. Table 1 is unclear as the best results are not consistently highlighted, and some indices show inferior performance compared to a random baseline, which is confusing.
3. The concept of new leakage risks introduced by the authors lacks substantial support from prior studies. The paper fails to clarify how these risks are quantified and mitigated, particularly the risk of intent leakage, which is critical for understanding the effectiveness of CodeCloak.
4. The criteria for code leakage are vague. The paper should clarify how it measures whether the intent behind the code has been leaked, considering that an adversary's ability to reconstruct the intent could still pose significant risks.

Minor Comments:
There are several typographical errors that need correction to enhance clarity and professionalism. For instance, the term "deep learning learning" should be corrected to "deep reinforcement learning."

**Questions:**

See the weaknesses section.

---

### Meta-Review · Area_Chair_jsdw · 2024-12-22

**Metareview:**

This paper introduces a novel LLM agent designed to mitigate code leakage by LLM code assistants. While the reviewers find the work interesting and recognize its potential, several concerns have been raised. The evaluation metrics and performance criteria are not adequately defined, making it difficult to assess the effectiveness of the proposed approach. Additionally, the paper suffers from certain presentation issues and lacks sufficient technical rigor. The authors are encouraged to improve the evaluation methodology, address the presentation problems, and strengthen the technical details. With these revisions, the work could be more competitive for future submission.

**Additional Comments On Reviewer Discussion:**

While the reviewers find the work interesting and recognize its potential, several concerns have been raised. The evaluation metrics and performance criteria are not adequately defined, making it difficult to assess the effectiveness of the proposed approach. Additionally, the paper suffers from certain presentation issues and lacks sufficient technical rigor. Some of these issues have been addressed during the rebuttal period, whereas some require more efforts to revise.

---

### Decision · Program_Chairs · 2025-01-22

Reject